# The uptake of key Essential Nutrition Action (ENA) messages and its predictors among mothers of children aged 6–24 months in Southern Ethiopia, 2021: A community-based crossectional study

**Aklilu Habte**[ID]*, **Addisalem Gizachew, Tekle Ejajo, Fitsum Endale**

School of Public Health, College of Medicine and Health Sciences, Wachemo University, Hosanna, Southern Ethiopia

* akliluhabte57@gmail.com

**Data Availability Statement:** All relevant data are within the manuscript and its Supporting Information files.

## Abstract

### Background

Essential nutrition action(ENA) is a framework for managing advocacy, establishing a foundation, and implementing a comprehensive package of preventive nutritional activities. Essential Nutrition Actions study studies provide current information on each nutrition action, allowing health systems to focus more on nutrition, which is critical in tackling the "double burden" of malnutrition: underweight and overweight. Hence, this study aimed at assessing the level of ENA practice and its predictors among mothers of children aged 6 to 24 months in southern Ethiopia.

### Methods

A community-based cross-sectional study was conducted from May 1 to 30, 2021 among randomly selected 633 mothers of children aged 6–24 months. A multi-stage sampling technique was used to access study participants. Data were collected by using a pretested, structured interviewer-administered questionnaire. To identify predictors of ENA practice, bivariable and multivariable logistic regression were used. The strength of the association was measured using an adjusted odds ratio with 95 percent confidence intervals. The statistical significance was declared at a p-value less than 0.05.

### Results

A total of 624 participants took part in the study, with a response rate of 98.6%. The uptake of key ENA messages among mothers was measured using 27 items, and it was found to be 47.4% (95% CI: 43.8, 51.4). Complementary feeding was the commonest ENA message practiced by 66.7% of respondents, while prevention of iodine deficiency disorder was practiced by only 33.7% of respondents. Variables namely, mother's education level of college and above [AOR: 3.90, 95% CI: 1.79, 8.51], institutional delivery [AOR: 2.75, 95% CI: 1.17,6.49], having PNC service [AOR: 2.95, 95% CI: 1.91, 4.57], being knowledgeable on

**Funding:** The author(s) received no specific funding for this work.

**Competing interests:** The authors have declared that no competing interests exist.

**Abbreviations:** AOR, Adjusted Odds Ratio; CF, Complementary Feeding; EBF, Exclusive Breast Feeding; EDHS, Ethiopian Demographic Health Survey; ENA, Essential Nutrition Actions; IDA, Iron Deficiency Anemia; IDD, iodine deficiency disorders; PCA, Principal Component Analysis; SDG, Sustainable Development Goal; SPSS, Statistical product and service solutions; VAD, vitamin A deficiency; WHO, World Health Organization.

ENA message [AOR: 2.37, 95% CI: 1.81, 3.26] and being a model household [AOR: 3.83,95% CI: 2.58, 5.69] were positively associated with a good uptake of key ENA messages. On the other hand, primiparity [AOR: 0.32, 95% CI: 0.21,0.56] was identified as a negative predictor.

## Conclusion

The overall practice of key Essential nutrition action messages in the study area was low as compared to studies. Stakeholders must step up their efforts to improve and hasten the utilization of maternal and child health services, especially institutional delivery and Postnatal care by focusing on uneducated women to promote compliance to key ENA messages. Furthermore, health workers need to focus on awareness-raising and model household creation.

## Introduction

At various phases of life, improving women's and children's nutrition necessitates a variety of activities, programs, and strategies [1]. The right to health is deeply rooted in good nutrition, which is enshrined in Article 25 of the Universal Declaration of Human Rights [2, 3]. Appropriate nutrition throughout the first 1,000 days (from the start of a mother's pregnancy to her child's second birthday) can have a significant impact on a child's capacity to grow, learn, and climb out of poverty [4–6]. Children who get the right diet during this crucial period are 10 times more likely to overcome life-threatening childhood diseases and have their own healthy families [7].

Essential nutrition action (ENA) is a framework for advocacy, foundation, and implementation of a comprehensive set of preventive nutritional interventions. The framework addresses seven key areas, covering appropriate child feeding, nutrition during pregnancy and lactation, and micronutrient supplementation [1, 8]. Those key messages can help us create a world where children are free of all forms of malnutrition, and they are delivered to women and children through various channels [1]. It is vital to break the intergenerational malnutrition cycle by implementing the seven key nutrition practices of the ENA framework within the first 1000 days [9]. However, more than half of the world's children are denied access to such life-saving interventions [7].

According to large-scale data, maternal and child malnutrition is the cause of 45 percent of under-five deaths [10]. In addition, it raises the risk of low birth weight, death, illness, infection, and delayed physical and cognitive development in undernourished children [1, 11]. Stunting has a long-term effect, and there is little room for catch-up development [12]. When considered on a national basis, these inequalities produced by stunting can result in lower school outcomes, and years on average in school, and harm a country's development and growth [13, 14].

According to the 2016 Ethiopia Demographic Health Survey (EDHS), 38 percent of children under the age of five are stunted, with 18 percent severely stunted [15]. Even though the Ethiopian government developed and implemented the Infant and Young Child Feeding (IYCF) Guideline at the national level, the practice keeps falling close to the WHO standard [16], with only 58% of infants being exclusively breastfed and only 7% of children receiving good possible complementary feeding. Even this, 9% of children under the age of six months use a feeding bottle [15]. Ethiopia has the highest rate of micronutrient deficiency among African countries, with 39.9% of children receiving inadequate iodine [17]. Only 23.3% of household units consume appropriate iodized salt [18]. Anemia affects one out of every six

Ethiopian women of reproductive age (17%) [19, 20]. Furthermore, in Ethiopia, a nationwide prevalence of subclinical vitamin A deficiency was found to be 33.9% in children aged 6 to 24 months [21].

Adequate implementation of seven key nutrition action messages can reduce undernutrition-related morbidity and mortality by nearly a quarter(25%) [7]. Inadequate IYCF interventions increase the risk of illness and mortality for infants and children by up to five times [22]. When pregnant women are given a well-adjusted diet, the likelihood of having children of small gestational age (SGA) drops by about 31% [23, 24]. To reduce death rates over the next five years, the WHO has set a goal of reducing anemia in women of reproductive age by 50% and reducing low birth rates by 30% by providing adequate nutrition throughout pregnancy [1, 7]. WHO is motivating healthcare providers to ensure that appropriate nutrition is provided at each stage of a human's life after the release of a new report, and this transformed investment in nutrition might save 3.7 million lives by 2025 [1].

Essential Nutrition Actions study reports provide current information on each nutrition action, allowing health systems to focus more on nutrition, which is critical in tackling the "double burden" of malnutrition: underweight and overweight. In addition, it will allow countries to include nutrition interventions in their national health and development strategies [15, 25]. Although the Ethiopian Federal Ministry of Health approved the ENA program as a child survival strategy, the level of practice and its predictors have not been thoroughly investigated, except for a single study conducted in northern Ethiopia [26]. Almost all of the studies focused solely on individual components, especially on Exclusive Breastfeeding and complementary feeding, which may not provide a comprehensive picture of ENA implementation. Knowing the overall practice of key ENA messages in the community is important for addressing malnutrition by identifying and intervening on components with low practice. Hence this study aimed at assessing the overall uptake of key ENA messages among mothers of children aged 6–24 months in the Lemo Distic in southern Ethiopia.

## Materials and methods

### Study area, design, and period

A community-based cross-sectional study was conducted in the Lemo District from May 1 to 30, 2021. The district is located 232 kilometers from Addis Ababa, Ethiopia's capital city, and 15 kilometers from Hossana, the Hadiya zone's capital. The total population in the district, based on the 2007 Central Statistics Agency estimation, is 118,594 (Male = 58,666 and Female = 59,928). There are a total of 35 kebeles in the district, (*Kebele*: *the smallest administrative unit in the current Ethiopian government structure under the district*). The primary health care units offering maternal and child health services were five health centers, one non-profitable non-governmental clinic, and 35 health posts (one in each kebele).

### Populations of the study

The source populations were all mothers of children aged 6 to 24 months in the Lemo district. Mothers with children aged 6–24 months in the selected kebeles of the district constituted the study population. Mothers who resided in the study area for at least six months were included, whereas those who were extremely ill during the data collection period were excluded.

### Sample size determination

The study sample size for the study was determined by using the single population proportion formula via the StatCalc menu of Epi-info version 7. The following parameters were used:

estimated prevalence of ENA of 46.5%, taken from a similar study conducted in northern Ethiopia [26], a 95% confidence interval, 5 percent degree of precision, design effect of 1.5, and non-response rate of 10%. The final sample size for the study was 633.

## Sampling procedures

A multi-stage sampling procedure was used to access study participants. Of 35 kebeles, fourteen were selected using a simple random sampling technique. Using the registration logbook of health extension workers, households with mothers of children aged 6–24 months were identified. Codes/numbers were then assigned to those houses that had eligible study participants, and a sampling frame was formed. The required sample size for each kebele was allocated by using a proportional allocation. By using a computer-generated random number study participants were selected and interviewed. Re-visits were made three times if selected respondents were difficult to access at the time of the survey.

## Data collection tools, methods, and personnel

A pretested, interviewer-administered questionnaire was used for the data collection. Eight diploma nurses with prior data collection experience conduct the data collection under the supervision of four public health officers. The data collection tool was developed by using the 2013 WHO Guideline for Essential Nutrition Actions, Enhancing Health and Nutrition for Maternal, Baby, Infant, and Young Children, Food and Agriculture Organization (FAO), and related literature [1, 7, 26, 27]. It was designed in the way to collect data on socio-demographic/economic, obstetric, and health system-related characteristics, exclusive breastfeeding, complementary feeding, sick child feeding, nutrition during pregnancy and lactation, vitamin A deficiency prevention, anemia prevention, and iodine deficiency prevention. Supervisors and data collectors were guided to sampling women's homes by the local Health Development Army (HDA) and community volunteers in each Kebele, and the respondents were then interviewed at their residential homes.

## Data quality management

Data collectors and supervisors got a one-day intensive training on data collection methods and procedures. The data collection tool was prepared in English, then translated into the local language (*Amharic*) by experts in that language, and finally back-translated to English to ensure that it fit the original meaning. A pre-test was conducted in the Soro district one week before the actual data collection for 5% of the sample size (29 women). To avoid any confusion, all necessary modifications were made based on the pre-test results. Supervisors and investigators closely oversaw the data collection processes daily to ensure the quality of the data. Investigators checked for missing values, inconsistencies, and outliers, and the possible corrections were made during the data collection period. Study participants were interviewed in private to reduce social desirability bias. To minimize the likelihood of recall bias respondents were given as much time as they needed for a good recall of long-term memories. In addition, inquiries were made, following an ordered sequence of events—starting with the present and thinking back to a point in time to cope with the recall bias.

## Data analysis

Epi-data version 3.1 was used to enter the data, which was then exported to SPSS version 23.0 for analysis. Inconsistencies and missing values were examined using running frequencies. Frequency distributions, mean, and standard deviation have all been computed as descriptive

statistics. The wealth status of households was determined using principal component analysis (PCA). Initially, 29 items were used and categorized into six categories: household property, livestock ownership, crop production in quintals, average monthly estimated income, agricultural land in hectares, and housing conditions [15]. PCA assumptions for sampling adequacy of individual variables were confirmed, including overall sampling adequacy calculation (KMO>0.6), anti-image correlations (> 0.4), and Bartlett Sphericity Test (p-value 0.05). Finally, three components were selected from the PCA, and the first component accounting for the maximum variation (48.9%) was used to classify the study participants' wealth status into quintiles [15].

To examine the relationship between outcome and explanatory variables, bivariable and multivariable logistic regression were conducted. Explanatory variables with a p-value <0.25 in the bivariable analysis were simultaneously moved into a multivariable logistic regression model to control for potential confounders. Those variables with a p-value< 0.05 in the final model were identified as determinants of ENA practice. Finally, the regression analysis findings have been reported using their adjusted odds ratios and the corresponding 95% confidence interval. Reports were presented in the form of charts, graphs, and figures. The statistical significance was declared at a p-value less than 0.05. The model fitness was assessed by using Hosmer and Lemeshow test, and the P-value was 0.521, indicating that the model provided the best fit. The level of multicollinearity was also examined using standard error cut-off two, but no multicollinearity was found among the variables studied.

## Definition and operationalization of variables of the study

**Dependent variable.**   Essential Nutrition Action (ENA): is an integrated preventive nutrition package comprising seven core components namely; exclusive breastfeeding, complementary feeding, sick children feeding, nutrition for women during pregnancy and breastfeeding, vitamin A deficiency prevention, anemia prevention, and iodine deficiency prevention [5, 7, 28]. A total of 27 items were used to assess ENA practice: exclusive breastfeeding(6 items), complementary feeding(5 items), sick child feeding(4 items), nutrition for women during pregnancy and breastfeeding(3 items), prevention of vitamin A deficiency(3 items), prevention of anemia(3 items), and prevention of iodine deficiency(3 items) [1, 7, 26, 27]. Information on these items was derived from the response to the questions like: 'Did you give sugar water, water, or butter, before breast during the first days of the baby's life?' For each practice assessment question, response categories were formed as '1 = for correct response' and '0 = for incorrect'. A composite index of ENA practice was computed, with the lowest value of zero indicating that women did not practice any ENAs and the highest value of 27 indicating that those women practiced all ENAs. Those who scored at or above the mean were considered to have a good practice, while those who scored below the mean were considered to have poor practice [26].

Exclusive breastfeeding: If a child under the age of six months consumes only breast milk and no other food, water, or other liquids (except medicines and vitamins, if necessary) [11, 26].

Complementary feeding: introduction of additional solid or semi-solid foods starting at 6 months, along with breast milk [7, 29].

**Explanatory variables.**   Household wealth index: Based on data from household assets and equipment, PCA was used to create a composite measure of respondents' wealth status. Finally, the first factor, which explained the maximum variation, was divided into quintiles [15].

Knowledge of ENA: Women who attained at least the mean score for the ENA knowledge assessment questions were labeled as knowledgeable, while those who did not were labeled as not knowledgeable [26].

Perceived distance to the nearest health facility: The distance between the mothers' residential home and the health facilities was measured in walking hours. This was categorized as 'closer' if mothers reported walking times of less than 30 minutes to reach the nearest health facility; otherwise, it was categorized as 'far' [30].

Being a model household (MHH): Those who have completed 75% of the four components of the health extension packages (HEPs) and have been certified [31]. These HEPs include family health (Maternal and Child Health), Infectious disease prevention and control (TB, HIV/AIDS, STIs, and Malaria), hygiene and environmental sanitation, and health education and communication [32].

Autonomy in household decision-making: A woman was said to be autonomous in decision-making if she made decisions independently or in collaboration with her husband. She was termed non-autonomous if she didn't make the decision herself or with the will of a third party [15, 25].

## Ethical approval and consent to participate

Ethical clearance was obtained from the Ethical Review Committee of Wachemo University School of Public Health, College of Medicine and Health Science with Rference number of WCU/121/2013. Before the study, all subjects provided their written informed consent. Before the study, informed written consent was taken from the study participants. The Lemo District Health Office also gave an official letter of cooperation. For those respondents under the age of 18 years, assent was obtained from their parents or guardian using standard disclosure procedures. The names of the respondents were kept confidential.

## Results

### Socio-demographic characteristics of respondents

Six hundred and twenty-four mothers took part in the study, with a response rate of 98.6%. The mean (±SD) age of mothers was 28.4(±5.3) years, and 182(29.3%) of them were between the age groups 20 and 24 years. In terms of education level, 230 (36.7%) of respondents had never had a formal education, and nearly a quarter, 149 (23.9%), had only completed primary school. Nearly two-thirds of the respondents, 407 (65.2%), were protestant by religion. In terms of their family size, nearly three quarters, 460(73.7%) had a maximum of 5 family members (Table 1).

### Maternal and child health service-related characteristics

Nearly half, 293(47.0%) of the mothers were multiparous. Five hundred fifty-six (89.1%) mothers received at least one ANC follow-up during their last pregnancy. Of those mothers who got ANC, 263 (47.3%) received four or more visits. More than two-thirds, 426 (68.3%) of respondents received postnatal care within the first six weeks of delivery. regarding the place of delivery, 573 (91.8%) mothers had their last child at the health facility. 439(70.4%) respondents were autonomous in decision-making (Table 2).

### Knowledge of respondents on key ENA messages

According to a composite knowledge measure for key ENA messages, more than half, 336 (53.8%) of the mothers were knowledgeable. Regarding the individual item, 63.4% and 58.1% of respondents knew about exclusive breastfeeding and complementary feeding, respectively. Prevention mechanisms for iodine deficiency were known by a relatively smaller proportion of respondents (36.6%) (**Fig 1**).

**Table 1. Socio-demographic characteristics of mothers of children aged 6–24 months old in Lemo District, Southern Ethiopia, 2021.**

| Variables Categories | Count | Percent |
|---|---|---|
| **Age (n = 624)** | | |
| <20 | 21 | 3.5 |
| 20–24 | 183 | 29.3 |
| 25–29 | 165 | 26.4 |
| 30–34 | 160 | 25.6 |
| 35+ | 95 | 15.2 |
| **Marital status (n = 624)** | | |
| Married | 562 | 90.1 |
| Divorced | 21 | 3.4 |
| Widowed | 13 | 2.1 |
| Single/never married | 28 | 4.4 |
| **Ethnicity (n = 624)** | | |
| Guraghe | 595 | 95.4 |
| Amhara | 19 | 3.0 |
| Others* | 10 | 1.6 |
| **Religion(n = 624)** | | |
| Protestant | 407 | 65.2 |
| Orthodox | 105 | 16.8 |
| Muslim | 99 | 15.9 |
| Catholic | 13 | 2.1 |
| **Wealth index (n = 624)** | | |
| Lowest | 124 | 19.9 |
| Second | 128 | 20.5 |
| Middle | 122 | 19.6 |
| Fourth | 123 | 19.7 |
| Highest | 127 | 20.4 |
| **Age of the child (n = 624)** | | |
| 6–12 months | 249 | 39.9 |
| 13–18 months | 272 | 43.6 |
| 19–23 months | 103 | 16.5 |
| **Mother's education level (n = 624)** | | |
| No formal education | 230 | 36.9 |
| Can read and write | 110 | 17.6 |
| Primary education | 149 | 23.9 |
| Secondary education | 85 | 13.6 |
| College and above | 50 | 8.0 |
| **Mother's Occupation(n = 624)** | | |
| Housewife | 284 | 45.5 |
| Unemployed | 179 | 28.7 |
| Private business work | 119 | 19.1 |
| Government employer | 42 | 6.7 |
| **Husband's educational level (n = 589)** | | |
| No formal education | 129 | 21.9 |
| Can read and write | 164 | 27.8 |
| Primary education | 169 | 28.7 |
| Secondary education | 78 | 13.1 |

*(Continued)*

**Table 1.** (Continued)

| Variables Categories | Count | Percent |
|---|---|---|
| College and above | 50 | 8.5 |
| **Family size (n = 624)** | | |
| ≤5 | 460 | 73.7 |
| ≥5 | 164 | 26.3 |
| **Husband's occupation (n = 589)** | | |
| Private business work | 185 | 31.4 |
| Farmer | 326 | 55.3 |
| Government employer | 44 | 7.5 |
| Daily laborer | 34 | 5.8 |
| **Sex of the child(n = 624)** | | |
| Male | 270 | 43.3 |
| Female | 354 | 57.7 |

**Exclusive breastfeeding (EBF) practice.** Almost all mothers, 622(99.7%) breastfed their children. More than three-quarters, 484(77.8%) of mothers practiced exclusive breastfeeding for the first six months, and more than half, 328(52.7%) began breastfeeding within one hour of delivery. Pre-lacteal feeding was practiced by 128(20.6%) respondents, with which butter was the commonest pre-lacteal food by 68 (53.1%) mothers (Table 3).

**Complementary feeding practice.** Four hundred forty-seven (70.7%) mothers, began complementary feeding for their infants at the end of six months. Over a fifth of respondents (20.7%) started complementary feeding before the sixth month, and just a small percentage, 54 (8.6%) started after the sixth month. In terms of minimum meal frequency, 320 (51.3%) of mothers fed their children supplementary foods at least three times a day, whereas 304 (48.7%) fed their children complementary foods as needed. Two hundred and eighty mothers (44.9%) used a variety of food products when preparing supplemental foods (i.e. animal-source foods, pulses and nuts, fruits, and vegetables rich in vitamin A). Almost three-quarters of respondents (73.8%) used at least one method to ensure the safety of complementary foods. The most common safety measure taken was proper handwashing before preparing and eating food (162,35.1%), followed by keeping the cleanliness of food preparation and serving utensils,115 (24.9%), and healthy storage and immediate serving of meals after preparation, 149(32.3%). The use of clean cups and bowls, 90(19.5%), and avoiding the use of difficult-to-keep-hygienic feeding bottles 63(10.1%) were not widely implemented safety precautions among mothers.

**The practice of feeding a child during and after illness.** One hundred Ninety-one (30.6%) mothers reported their child had been sick in the past 14 days. 115 (60.2%), 93 (48.5%), and 102 (53.4%) mothers of sick children had provided more than the usual amount of breastfeeding, food, and fluid, respectively(Fig 2).

**Mother's practice on nutrition during pregnancy and lactation.** Only 243 (38.9%) mothers acquired a variety of foods for the period of pregnancy and breastfeeding, especially animal products, fruits, and vegetables, and nearly six out of ten (357, 57.2%) mothers consumed one additional meal (snack) per day (Table 4).

**The practice of vitamin A deficiency prevention by mothers (VAD).** Over three-fifths of mothers (60.9%) did not receive Vitamin A supplements within 45 days of delivery. Only 228 (36.5%) of children received vitamin A supplementation twice a year. More than half, 339 (54.3%) have eaten vitamin A-rich foods as part of their meal in the last 24 hours. Green vegetables accounted for the most food item consumed(29.2%), followed by fruits 79(12.7%), and animal products 59(9.4%).

**Table 2. Maternal and child health service utilization patterns of mothers of children aged 6–24 months old in Lemo District, Southern Ethiopia, 2021.**

| Variables Category | Frequency | Percent |
|---|---|---|
| Parity (n = 624) | | |
| Primiparous | 125 | 20.0 |
| Multiparous | 293 | 47.0 |
| Grand multiparous | 206 | 33.0 |
| Planning status of last pregnancy (n = 624) | | |
| Planned | 454 | 72.8 |
| Mistimed | 108 | 17.3 |
| Unwanted | 62 | 9.9 |
| Frequency of ANC visits(n = 624) | | |
| No | 68 | 10.9 |
| 1 visit | 69 | 11.1 |
| 2–3 Visits | 224 | 35.9 |
| ≥4 visits | 263 | 42.1 |
| Place of delivery (n = 624) | | |
| Health center | 427 | 68.4 |
| Hospital | 146 | 23.4 |
| Home | 51 | 8.2 |
| Mode of delivery (n = 624) | | |
| SVD | 528 | 84.6 |
| Instrumental delivery | 62 | 9.9 |
| Caesarean delivery | 34 | 5.5 |
| Having PNC visit (n = 624) | | |
| Yes | 426 | 68.3 |
| No | 198 | 31.7 |
| Time spent to reach nearby health facility (n = 624) | | |
| ≤30min | 193 | 30.9 |
| >30min | 431 | 69.1 |
| Means of transportation(n = 624) | | |
| On foot | 452 | 72.4 |
| By Bajaj/Taxi | 172 | 27.6 |
| Being a model household(n = 624) | | |
| Yes | 325 | 52.1 |
| No | 299 | 47.9 |
| Autonomy in decision making (n = 624) | | |
| Autonomous | 439 | 70.4 |
| Non-autonomous | 185 | 29.6 |

**Mothers' practice of preventing iron deficiency anemia (IDA).** Four hundred thirty-nine (70.4%) and 221(35.4%) mothers obtained iron-folic acid during their recent pregnancy and lactation respectively. During the last pregnancy, only one-third of women (32.9%) mothers received food items rich in iron (meat, liver, kidney, and heart) and animal products a minimum once a day. Concerning deworming, Just 287(45.9%) and 134(21.4%) of respondents received anti-helminths during pregnancy and lactation, respectively.

**The practice of respondents on the prevention of iodine deficiency disorders (IDDs).** More than half, 368(59.1%) of respondents had Iodized salt in their homes. Regarding the

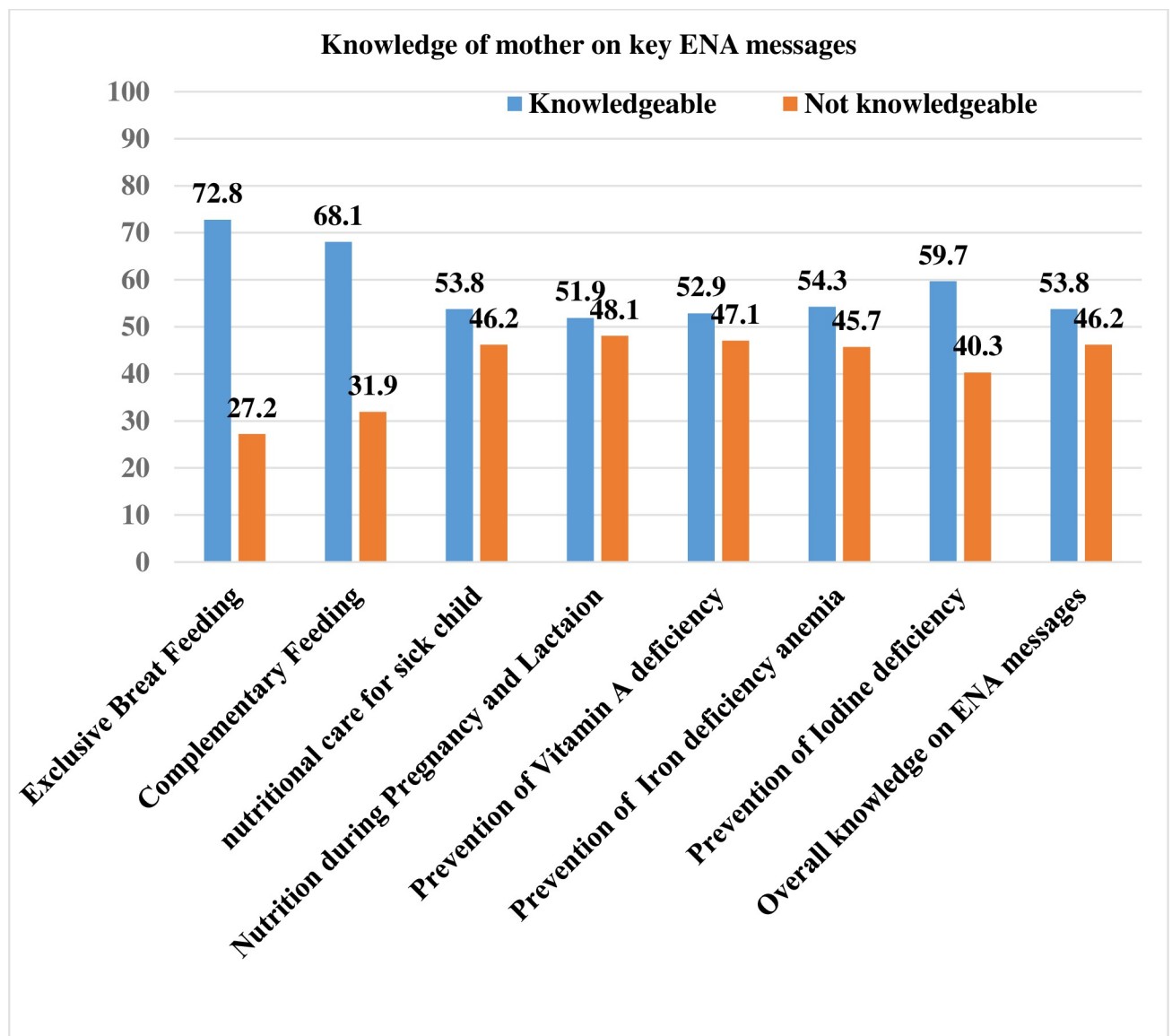

**Fig 1. Knowledge of mothers towards key ENA messages in Lemo District, Southern Ethiopia, 2021.**

timing, only 207(33.5%) of respondents add salt at the end of the cooking process. Whereas 327(52.4%) and 88(14.1%) added salt in the middle and beginning of cooking, respectively. Less than half (275, 44.1%) of mothers store salt in a dim and closed jar.

## The uptake of key ENA messages among mothers

The overall uptake of key ENA messages among mothers was measured using 27 items, and 47.4% (95% CI: 43.8, 51.4) had a good uptake of key ENA messages. Complementary feeding and exclusive breastfeeding were the most commonly practiced item by 66.7% and 62.8% of respondents, respectively. On the other hand, only one-third(33.7%) of respondents had a good practice in the prevention of iodine deficiency disorder (**Fig 3**).

**Table 3. Exclusive breastfeeding practice among mothers of children aged 6–24 months of age in Lemo District, Southern Ethiopia, 2021.**

| Practice assessment variables | Frequency (%) |
|---|---|
| **Have you ever breastfed your child? (n = 624)** | |
| Yes | 622(99.7) |
| No | 2(0.3) |
| **How long do you give exclusively breast milk (n = 622)?** | |
| Less than six month | 63(10.1) |
| Six month | 484(77.8) |
| More than six months | 75(12.1) |
| **Breastfeeding initiation time (n = 622)** | |
| Immediately after birth (within 1 Hr) | 328(52.7) |
| Hours after birth | 131(21.1) |
| Days after birth | 85(13.7) |
| I don't remember | 78(12.5) |
| **On the first day of the baby's life, pre-lacteal feeding was provided(n = 622)** | |
| Yes | 38(6.1) |
| No | 584(93.9) |
| **Pre-lacteal feeding items(n = 128)** | |
| Sugar water | 38(29.7) |
| Water | 43(33.6) |
| Butter | 68(53.1) |
| **Squeeze out the first milk (colostrum) and throw it away (n = 622)** | |
| Yes | 135(21.7) |
| No | 487(78.3) |
| **In the first six months, the child took something from a bottle other than breast milk (n = 624)** | |
| Yes | 139(22.3) |
| No | 483(77.6) |
| **Breastfeeding frequency per day? (n = 624)** | |
| 10 times and more | 254(40.7) |
| less than 10 times | 270(43.3) |
| I don't remember | 100(16.0) |

## Predictors of practice toward key ENA messages

Six variables were identified as important determinants of good ENA practice in multivariable logistic regression analysis. The current study identified that there was a significant association between educational level and the practice of key ENA messages. In comparison to mothers with no formal education, mothers who had completed college or higher were 3.9 times more likely to practice essential ENA messages [AOR: 3.90, 95% CI: 1.79, 8.51]. In terms of parity, primiparous women were 68% less likely to have a good uptake of key ENA messages as compared to multiparous mothers [AOR: 0.32, 95% CI: 0.21,0.56]. Compared to mothers who gave birth at home, mothers who gave birth at a health facility were 2.75 times more likely to implement key ENA messages [AOR: 2.75, CI: 1.17,6.49]. Receiving postnatal care was also found to be an important predictor of using key ENA messages. Mothers who received postnatal care were 2.95 times more likely to have a good practice of key ENA messages [AOR: 2.95, CI: 1.91, 4.57]. The odds of good uptake of key ENA messages were 2.34 times greater among mothers with good knowledge of key ENA messages as compared to those mothers with poor knowledge [AOR: 2.37, CI: 1.81, 3.26]. Finally, the research found that living in a model household

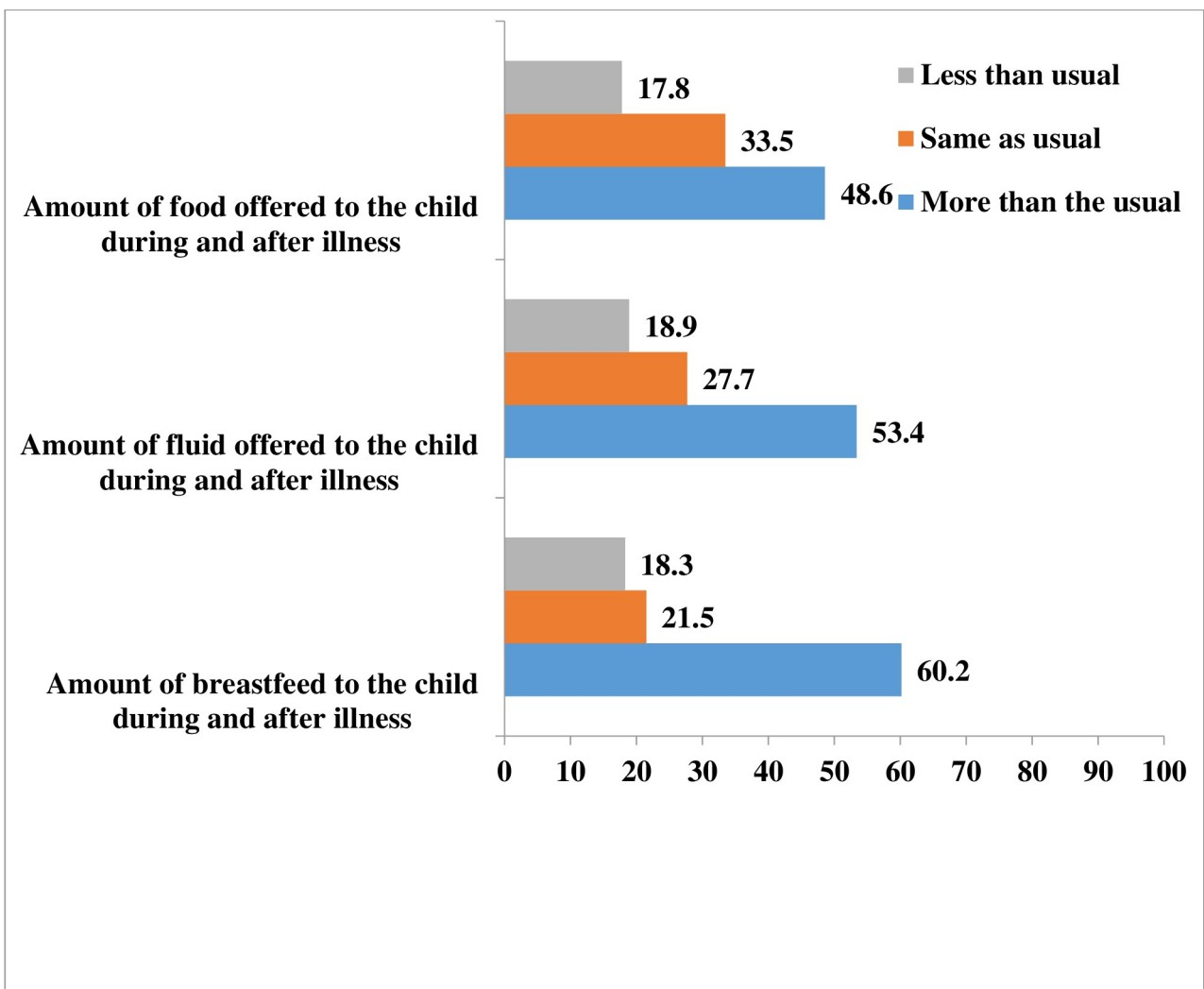

**Fig 2. The practice of mothers on feeding sick children during and after an illness in Lemo District, Southern Ethiopia, 2021.**

increases the likelihood of implementing key ENA messages. Mothers from model households were 3.83 times more likely than their counterparts to have a good practice [AOR: 3.83, CI: 2.58, 5.69] (Table 5).

## Discussion

This study aimed at assessing the uptake of key ENA messages and its predictors among mothers of children aged 6 to 24 months. In the study area, 47.4% of mothers had a good practice of implementing key ENA messages, which is comparable to a study conducted in northern Ethiopia, where 46.5% of mothers complied well with key ENA messages [26]. Good practice of EBF in the current study area was 62.8% which is almost comparable with the national average of Mini EDHS 2019 (59%) and other studies conducted in Turkey (59.8%) and Western Ethiopia [33–35]. The practice, however, is lower than several studies conducted in Pakistan (72%), Eastern Ethiopia (81.1%), Central Ethiopia (75.7%), southern and northern Ethiopia (88.0%), and northern Ethiopia (84%) [36–40]. This could be due to disparity in the study area, as most of these studies with higher figures were conducted in urban populations. On the other hand,

**Table 4. Pattern nutrition during pregnancy and lactation among mothers of children aged 6–24 months in Lemo District, Southern Ethiopia, 2021.**

| Assessment of Dietary Practice during pregnancy and lactation | Frequency (%) |
|---|---|
| Eats one additional meal every day during pregnancy and lactation (n = 624) | |
| yes | 357(57.2) |
| no | 267(42.8) |
| Most typical meal patterns during Pregnancy and/or lactation in 24hr | |
| Breakfast- lunch- dinner | 267(42.8) |
| Breakfast-snack–lunch- dinner | 175(28.0) |
| Breakfast- lunch-snack- dinner | 121(19.4) |
| Breakfast-snack- lunch-snack-dinner | 61(9.8) |
| Dietary diversity during pregnancy and lactation(n = 624) | |
| Did you eat a variety of foods, particularly animal products (meat, milk, eggs, etc.), plus fruits & vegetables during pregnancy and lactation? | 243(38.9) |
| A starchy food(millet, maize, rice, wheat, or Teff, Oats, Carrots, or sweet potatoes that are yellow or orange inside) | 377(60.4) |
| Dark green leafy vegetables (Such as kale, cabbage. . .) | 429(68.8) |
| Fruits & Vegetables (pumpkin, any ripe mangoes, ripe papayas, lemon, orange, banana. . .) | 142(22.7) |
| Organ meat (liver, kidney, heart, or other organ meats) | 111(17.7) |
| Legumes, nuts, and seeds (Dried beans, dried peas, lentils, nuts, seeds) | 324(51.9) |
| Milk and milk products (cheese, yogurt, milk, or other milk products) | 101(16.2) |
| Meat and fish (like beef, lamb, goat, chicken, and fish or other blood-based foods) | 83(13.3) |

EBF practice was higher than studies conducted in Saudi Arabia (24.4%), average estimates for the Eastern and Southern African region (47%), developing countries (39%), and Eastern Ethiopia (45.8%) [41–43]. The disparity may be attributable to the continued promotion of free delivery of maternal and child health care, resulting in a substantial rise in ANC follow-up and skilled delivery, particularly in the study area.

Regarding complementary feeding, 52.7% of mothers practiced proper complementary feeding. This finding is higher than studies carried out elsewhere [39, 44–46]. The disparities might be attributed to the study period and the engagement of governmental and non-governmental organizations that are encouraging the worth of complementary feeding all through mass media at the instant. Besides, health extension workers often make home-to-home visits to support families in accessing basic health extension packages such as home-based health education and promotion programs such as proper IYCF. However, this finding is lower than the WHO cut-off point (80 to 94%) for good practice of complementary feeding [47] and previous study findings elsewhere [26, 35, 48, 49]. This might be due to disparity in education, socioeconomic, and cultural conditions, as most of the higher figured studies were from developed countries. This illustrated that stakeholders need continuous efforts to enhance the practice of optimal complementary feeding by offering community nutrition education and consultation as one part of maternal and child health care services.

The study revealed that 44.9% of mothers practiced proper feeding (increasing the frequency of feeding during and after illness) for their sick child. The finding was greater than the study conducted in northern Ethiopia (38.0%) [26] and lower than a similar study conducted in central Ethiopia (53.6%) [50]. The variation may be due to differences in the study area where the latter was done among mothers residing in urban setups and they might be more conscious of the measures to be trailed during childhood ailment through mass media.

The likelihood of having a good practice of key ENA messages was higher among mothers who attend college and above as compared to mothers with no formal education. This finding

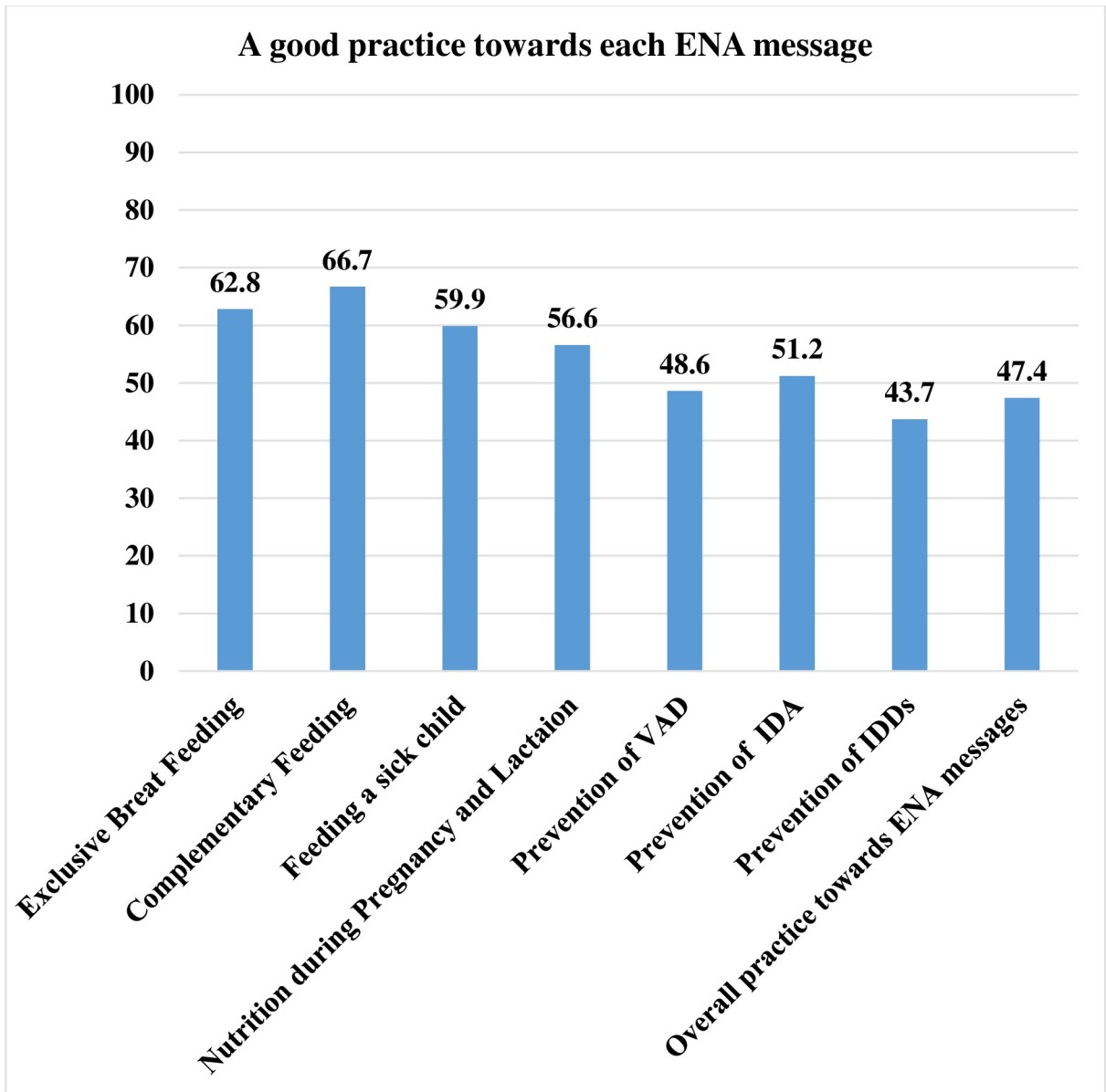

**Fig 3. Distribution of uptake of key ENA messages among mothers of children aged 6–24 months in Lemo District, Southern Ethiopia, 2021.**

was supported by studies conducted in Northern Ethiopia [26, 51]. This could be because uneducated women are less engaged with nutritional education than educated mothers. In addition, those uneducated mothers may also be less able to access and deal with information from different outlets, such as paperbacks, pamphlets, and magazines. Furthermore, more educated women might just have a better chance than their peers of accessing nutrition education through mass media and other sources.

Knowledge of mothers on key ENA messages was also identified as a significant predictor of ENA uptake. This was supported by studies conducted in northern Ethiopia [26, 52]. This implies that enhancing mothers' knowledge of key ENA messages is a crucial intervention of choice for implementing long-term strategies to increase effective ENA practices. In addition,

**Table 5. Determinants of the practice of key ENA messages among mothers of children aged 6–24 months in Lemo District, Southern Ethiopia, 2021.**

| Characteristics of respondents (n = 624) | Uptake of key ENA messages | | AOR(95%CI) | p-value* |
|---|---|---|---|---|
| | Good | Poor | | |
| **Age** | | | | |
| 35+ | 44(46.3) | 51(53.7) | 1 | |
| 30–34 | 70(43.7) | 90(56.3) | 1.12(0.58,2.18) | .737 |
| 25–29 | 81(49.1) | 84(50.9) | 1.24(0.62,2.49) | .541 |
| 20–24 | 89(48.6) | 94(51.4) | 1.44(0.75,2.79) | .275 |
| <20 | 12(57.1) | 9(42.9) | 2.66(0.64,6.10) | .179 |
| **Mother's education** | | | | |
| No education | 102(44.3) | 128(55.7) | 1 | |
| Can read and write | 31(28.2) | 79(71.8) | 0.69(0.39,1.24) | 0.220 |
| Primary education | 68(45.6) | 81(56.4) | 1.02(0.61,1.69) | 0.343 |
| Secondary education | 61(71.8) | 24(28.2) | **2.42(1.29,4.51)**[**] | 0.006 |
| College and above | 34(68.0) | 16(32.0) | **3.90(1.79,8.51)**[**] | 0.001 |
| **Mother's occupation** | | | | |
| Private business work | 55(46.2) | 64(53.8) | | |
| Government employer | 19(45.2) | 23(54.8) | 0.75(0.30,1.84) | 0.525 |
| Housewife | 124(43.7) | 160(56.3) | 0.75(0.44,1.29) | 0.305 |
| Un employed | 98(54.7) | 81(45.3) | 1.19(0.67,2.12) | 0.554 |
| **Husband educational** | | | | |
| No education | 54(40.9) | 78(59.1) | 1 | |
| Can read and write | 75(45.7) | 89(54.3) | 1.40(0.79,2.47) | .238 |
| Primary education | 78(46.1) | 91(53.8) | 1.37(0.78,2.38) | .270 |
| Secondary education | 39(50.0) | 39(50.0) | 1.28(0.65,2.54) | .473 |
| College and above | 31(62.0) | 19(38.0) | 2.06(0.93,4.55) | .075 |
| **Wealth index** | | | | |
| Lowest | 50(40.3) | 74(59.7) | 1 | |
| Second | 61(47.6) | 67(52.3) | 1.13(0.61,2.08) | .501 |
| Middle | 56(45.9) | 66(54.1) | 1.58(0.84,2.99) | .157 |
| Fourth | 62(50.4) | 61(49.6) | 1.62(0.88,3.00) | .123 |
| Highest | 67(52.8) | 60(47.2) | 1.65(0.89,3.06) | .110 |
| **Family size** | | | | |
| >5 | 70(42.7) | 94(57.3) | 1 | |
| ≤5 | 226(49.1) | 234(50.9) | 1.08(0.66,1.75) | 0.612 |
| **Parity** | | | | |
| Grand multiparous | 123(59.7) | 83(40.3) | 1 | |
| Multiparous | 127(43.3) | 166(56.7) | 0.48(0.39,0.89) | .041 |
| Primiparous | 46(36.8) | 79(63.2) | **0.32(0.21,0.56)**[**] | .021 |
| **ANC visits** | | | | |
| No visits | 25(36.8) | 43(63.2) | 1 | |
| 1 visit | 26(37.7) | 43(62.3) | 0.76(0.33,1.75) | .520 |
| 2–3 visits | 100(44.6) | 124(55.4) | 1.20(0.61,2.35) | .595 |
| Four and more visits | 145(55.1) | 118(44.9) | 1.59(0.82,3.07) | .169 |
| **Place of delivery** | | | | |
| Home | 12(23.5) | 39(76.5) | 1 | |
| Health institution | 284(49.6) | 289(50.4) | **2.75(1.17,6.49)**[**] | 0.021 |
| **Getting PNC** | | | | |
| No | 58(29.3) | 140(70.7) | 1 | |

*(Continued)*

**Table 5.** (Continued)

| Characteristics of respondents (n = 624) | Uptake of key ENA messages | | AOR(95%CI) | p-value* |
|---|---|---|---|---|
| | Good | Poor | | |
| Yes | 238(55.9) | 188(44.1) | **2.95(1.91,4.57)**[**] | <0.001 |
| **Being a Model Household** | | | | |
| No | 87(29.1) | 212(70.9) | 1 | |
| Yes | 209(64.3) | 116(35.7) | **3.83(2.58,5.69)**[**] | <0.001 |
| **Knowledge of ENA** | | | | |
| Not knowledgeable | 130(45.5) | 156(54.5) | 1 | |
| Knowledgeable | 165(49.0) | 172(51.0) | 1.37(0.91,2.06) | 0.127 |
| **Decision making** | | | | |
| Non-autonomous | 74(4.0) | 111(60.0) | 1 | 1 |
| Autonomous | 222(50.6) | 217(49.4) | 1.24(0.81,1.91) | 0.323 |

**Key:** 1: Reference category

AOR = Adjusted odds ratio

COR = Crude odds ratio

*Statistically significant at p-value<0.25

** Statistically significant at p-value <0.05

a concerted effort is needed by health care providers through information education communication to achieve better compliance of mothers towards nutrition-based messages.

The uptake of key ENA messages was positively influenced by the place of delivery. This finding was in a tandem with studies carried out in Wereilu [26] and Azezo District [53], both in northern Ethiopia. This is justified by the increased likelihood of obtaining relevant maternal and child feeding information for mothers who gave birth in health facilities, which can tackle traditional beliefs that obstruct optimal postpartum child feeding.

Also, there was a significant association between the uptake of key ENA messages and postnatal service utilization which was in line with previous findings of the studies conducted elsewhere [26, 54, 55]. This might be due to the effect of information and education provided by healthcare providers during postnatal visits. Therefore, for better practice of key ENA messages, due emphasis is required from the concerned bodies to offer PNC service packages to mothers.

Moreover, the study revealed being primiparous was found to negatively influence the uptake of key ENA messages. this is supported by studies conducted in northern Ethiopia [26, 56], Vietnam [33], and Italy [36]. According to studies, even though primiparous mothers were more likely to initiate BF, they did so for a shorter period and introduced complementary foods earlier due to their lower self-confidence in their competence to breastfeed [33, 36]. Some findings also found that some breast problems, such as breast pain and infections, are common in primiparous women and are known to delay the early initiation of breastfeeding, resulting in a decrease in breastfeeding frequency [37]. All of this could result in the start of CF before six months, resulting in low uptake of Key ENA messages. Hence, primiparous mothers should be counseled on the importance of adhering to essential ENA messages during the prenatal and postpartum periods.

Finally, the study revealed that being a model household(MHH) was significantly associated with a good practice of key ENA messages. This might be because those mothers from non-model households may be less likely to have contact with health care providers and, as a result, may not have enough knowledge about those key ENA messages, potentially ending up with

low practice. On the other hand, through extensive preparation, encouragement, and follow-up and family education on MNCH programs for those chosen to be role models, HEWs invest more time in the capacity-building component for model HHs. This successive training, support, and follow-up could contribute to the growth of skills and make them exercise key ENA messages too well [57, 58].

There were both strengths and shortcomings in this research. The study is among a few of its kind to assess the level and predictors of the practice of key ENA messages comprehensively since almost all of the studies were focused on individual components. The results of the study may have significant policy implications for the further enhancement of ENA packages. Although the necessary efforts have been made to reduce the potential shortcomings of this study, readers should be careful when interpreting the results. Since the study was based on self-reports, the respondents might be prone to social desirability bias. Finally, because women were asking about incidents that had already occurred before the study period, there may be a risk of recall bias.

## Conclusion

The overall practice of key ENA messages in the study area was low as compared to previous studies. Variables namely maternal education of college and above, being primiparous, institutional delivery, having PNC service, having adequate knowledge of ENA, and being a model household were identified as significant predictors of ENA practice. Stakeholders must step up their efforts to improve and hasten the utilization of maternal neonatal and child services especially on skilled delivery and postpartum visits by emphasizing women with no formal education. Furthermore, health extension workers should concentrate on awareness-raising and model household creation to promote adherence to key ENA messages. Primiparous mothers should be counseled on the importance of adhering to essential ENA messages during the prenatal and postpartum periods.

## Supporting information

**S1 Dataset. Data set supporting the findings of the study.**
(SAV)

**S1 File. Data collection tool for the study.**
(DOCX)

## Acknowledgments

We are very grateful to the Department of Public Health of Wachemo University for all the support we have been provided to conduct this research. We are grateful for their contributions during the study to our data collectors, managers, and study participants.

## Author Contributions

**Conceptualization:** Aklilu Habte, Fitsum Endale.

**Data curation:** Aklilu Habte.

**Formal analysis:** Aklilu Habte, Tekle Ejajo.

**Investigation:** Aklilu Habte, Addisalem Gizachew, Tekle Ejajo.

**Methodology:** Aklilu Habte, Addisalem Gizachew, Tekle Ejajo, Fitsum Endale.

**Resources:** Aklilu Habte.

**Software:** Aklilu Habte.

**Supervision:** Addisalem Gizachew, Tekle Ejajo.

**Validation:** Aklilu Habte.

**Writing – original draft:** Aklilu Habte, Addisalem Gizachew.

**Writing – review & editing:** Aklilu Habte, Tekle Ejajo, Fitsum Endale.

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
