## [Decision Letter · Decision Letter 0]

15 Jun 2022

PONE-D-21-34897The Uptake of Key Essential Nutrition Action (ENA) Messages and Its Predictors among Mothers of Children from 6-24months in Southern Ethiopia, 2020: A Community Based Crossectional StudyPLOS ONE

Dear Dr. Habte,

Thank you for submitting your manuscript to PLOS ONE. After careful consideration, we feel that it has merit but does not fully meet PLOS ONE’s publication criteria as it currently stands. Therefore, we invite you to submit a revised version of the manuscript that addresses the points raised during the review process. Your manuscript has been assessed by two expert reviewers, whose comments are appended below. The reviewers have highlighted concerns about several aspects of the methodology and presentation which must be addressed for the manuscript to be suitable for publication. Please ensure you respond to each point carefully in your response to reviewers document, and modify your manuscript accordingly.

We look forward to receiving your revised manuscript.

Kind regards,

Joseph Donlan

Editorial Office

PLOS ONE

Journal Requirements:

Reviewers' comments:

Reviewer's Responses to Questions

**Comments to the Author**

1. Is the manuscript technically sound, and do the data support the conclusions?

Reviewer #1: Yes

Reviewer #2: Yes

2. Has the statistical analysis been performed appropriately and rigorously? 

Reviewer #1: Yes

Reviewer #2: Yes

3. Have the authors made all data underlying the findings in their manuscript fully available?

Reviewer #1: Yes

Reviewer #2: Yes

4. Is the manuscript presented in an intelligible fashion and written in standard English?

Reviewer #1: Yes

Reviewer #2: Yes

5. Review Comments to the Author

Reviewer #1: General comments

It is good work. There are a lot of grammar and punctuation error, give time for your document and try to modify them. Above those comments try to see critically your document for minor errors.

Specific comments

Title: Import hyphens (Community-based Cross-sectional)

Abstract:

• Be consistent throughout your document (either 6 to 24 months or 6 – 24 months, check with what you put in your title)

• The practice of mothers towards key ENA messages…

• --- the magnitude of good ENA practice was 296. (47.4%)—better to remove the figure and put only the percentage in the abstract

• Mother’s education level [AOR: 3.90, 95%CI: 1.79, 8.51], parity [AOR: 2.72, 95%CI: 1.51,4.89], place of delivery[AOR: 2.75, 95%CI: 1.17,6.49], having PNC service[AOR: 2.95, 95%CI: 1.91, 4.57] and being a model household [AOR: 3.83,95%CI: 2.58, 5.69] were identified as a significant determinants of a good practice---- Better if you put separately which are positively associated, and which are negatively associated.

• put a space between 95%CI---95% CI, a space between word and [

• The overall practice of key ENA messages in the study area was low…. Compared to what?

• The MNCH, SDS, PNC--- if possible try to avoid abbreviations in Abstract, unless you have to state them initially

• Key messages – better if you include your study area, and study participants

Background

• -----programs, and strategies[1]—Put a space between words and references across the whole document ( strategies [1])

• The key areas of ENA is in children between 6 – 23 months but your study is on 6 – 24 months, why?

• Why you use the report of EDHS 2016 ?

• Your statement of the problem states around, not exactly on the ENA components. And you did not clearly identify studies on ENA, and the limitations of each study, and what your study adds on top of those studies (the gap seems vague).

Methods and materials

• NGO ----state it

• The study populations were women from selected kebeles who fulfilled the eligibility criteria – this is totally wrong. Clearly put the difference between your source population and study population. Then you have to put the exclusion and inclusion criteria separately

• The sampling procedure you followed to get the study participants is not feasible in such community (Community which is dispersed randomly, and have no household number) … in such community it is better to use cluster sampling technique.

• Why nurses as data collector, and PH as supervisor? Why not other professionals too

• The variance inflation factor(VIF) >10 was used to check for multicollinearity between the explanatory variables---there is no space in logistic regression to run multicollinearity, this kind of test seems pseudo, better if you see the standard errors of each variable.

• Finally, the regression analysis’ findings have been reported using odds ratios and the corresponding 95 % confidence interval --- Adjusted odds ratios

• Did you check model fitness and model adequacy, Did you check outliers…it should be mentioned

• Ethical clearance was obtained from the Ethical Review Committee of Wachemo University School of Public Health, College of Medicine and Health Science—Put the IRB number

Result

• Where did you get such like Age classification of children ---- 6-11, 12-17, 18-23

• Family size classification –why you use cut of 5, what is the recommendation in Ethiopian context

• When it came to the number of children, 293(47.0%) of respondents were multiparous. Do you mean children are multiparous???

• Remove the words like the majority, minority, small percentage….. they have no common interpretation

• For numbers >99 it is advisable to put in number rather than stating in words

• The font size across your document should be similar

• Your discussion flow should be from your study finding, then to similar study settings, then to Africa, then to the world. But you didn’t follow coherence.

• Make your discussions short and smart, it seems the report of your and other findings

• Your justifications in the discussion should be supported by some evidence, even if your thought of thinking may be there but advisable to support with references

• Your conclusion should be strong enough based on standard’s

References

• Check the grey literature when you use ENDNOTE, they need manual arrangement in the software

Reviewer #2: Comments to the author

General comments:

Five predictors (maternal, household) for essential nutrition message for nutrition of children before 1000 days of life have been set up with the manuscript. Individual, maternal, household and health system factors need more focus for improving child nutrition and health by policy-makers.

The manuscript reads well with consistency between sections although the wordcount remains wide. I appreciated the background and methods sections, they look very informative and consistent with the title and the aim of the study. Some restructuration and editing need to be done in the results, discussion, and reference list for better understanding of the text. The discussion is long as the amount of data that the paper has gathered, analyzed and interpreted. Also, the limitations of the study seem unclear, and some contrasting results have to be clarified and detailed.

Specific comments:

Method and study design section:

• Sampling section:

o When two stages sampling methods for participants has been considered, the authors should clarify and explain the process. Kebele level with simple random selection, household level with random selection, and participants level with simple random selection look like three stages of sampling instead of two.

Results

• When the dates are mentioned in the methods section, the date doesn’t add value to the title of each table.

• Knowledge of respondent section: the sentence should start by a capital letter.

• confusing results come out from the association between being primiparous mothers and good ENA messages because of the free time to care for children and to cook healthy food for family. But this is a bit confusing and contrasting with other studies where being multiparous could enable mothers to keep more knowledge and experienced practices compared to primiparous mothers. How the authors can discuss these explanations?

• Multivariable table 5 shows the significant variables that are associated with good practices of key ENA in the study. Being educated (more than secondary level), birth at health facilities, prenatal care (more than four time). The authors should show the variables that have been removed from the adjusted models because of the collinearity.

• Limitations of the study and bias (recall bias and desirability bias) has been risen in the study. The authors should say how they attempt to mitigate this risk of bias?

References

• Check for the use of capital letters in titles, abbreviation for the journal, the consistency of page numbers in the references.

• The use of capital letter in the titles (either for the first letter of the title or first letter for each word) should be consistent for the whole list of references.

• The authors should edit the publication year for the reference number 3

• The consistency needs to be followed with the style of references, particularly the using of “et al”. For instance, this term should be used with consistency after the third authors when the list of authors includes more than six authors. See references number 6, 15,19, 25…

• The authors should edit the spelling of the WHO as author of the citation in the list of references: either writing “WHO”, or World Health Organization” instead of “Organization, WH”. See references number 1, 2, 8, 22, 26…

• The reference 49 needs to edit the page number.

6. PLOS authors have the option to publish the peer review history of their article (what does this mean?). If published, this will include your full peer review and any attached files.

Reviewer #1: **Yes: **Getahun Fentaw Mulaw

Reviewer #2: **Yes: **Christian Bwangandu Ngandu

---

## [Author Response · Author response to Decision Letter 0]

19 Jun 2022

The responses to reviewers have been attached as a "Response to Reviewers" in the submission system

---

## [Editor Report · Decision Letter 1]

4 Sep 2022

PONE-D-21-34897R1The Uptake of Key Essential Nutrition Action (ENA) Messages and Its Predictors among Mothers of Children aged 6-24 months in Southern Ethiopia, 2021: A Community-Based Crossectional StudyPLOS ONE

Dear Dr. Habte Hailegebireal,

Thank you for submitting your manuscript to PLOS ONE. After careful consideration, we feel that it has merit but does not fully meet PLOS ONE’s publication criteria as it currently stands. Therefore, we invite you to submit a revised version of the manuscript that addresses the points raised during the review process.

We look forward to receiving your revised manuscript.

Kind regards,

Saqlain Raza

Academic Editor

PLOS ONE

Journal Requirements:

Additional Editor Comments:

1. In Table 1, the variable 'age' has been categorized as categorical variable, and estimated its frequencies and percentages of age-groups. Secondly, in the results before Table 1, authors found mean and SD of age. This creates ambiguity of whether the variable 'age' has been considered as continuous of categorical. Generally, we take the variable 'age' as continuous variable. Authors need to follow the variable type and justify the analysis and results in the same way.

2. In Table 1, authors have estimated WIQs along with socioeconomic status of the participants. It is quite a bit confusing numbers in the frequency. Authors need to ensure the authenticity of the calculations done through PCA.

3. Standardized age-groups for children are 6-11 months, 12-17 months, and 18-23 months. 23 months means the child has completed 23 months or 2 years of age. Authors can adjust children according to standardized format.

4. In Table 3, only 63 children were not exclusively breastfed. In another variable on the same table, 139 children were given some food other than breast milk. The two variables contradicts each other. This is a question mark on the data quality. How would the authors justify this?

5. In Table 5, give only one of 'COR' or 'AOR' and the remove the other.

6. Include in your study some more relevant literature like:

https://journals.plos.org/plosone/article?id=10.1371/journal.pone.0263470

https://pubmed.ncbi.nlm.nih.gov/35211439/

7. In the abstract, methods section is unnecessarily lengthy. Authors need to shorten it.
---

## [Author Response · Author response to Decision Letter 1]

7 Sep 2022

The response to reviews have been attached as a "Response to academic editor" in the submission process

---

## [Editor Report · Decision Letter 2]

13 Sep 2022

The Uptake of Key Essential Nutrition Action (ENA) Messages and Its Predictors among Mothers of Children aged 6-24 months in Southern Ethiopia, 2021: A Community-Based Crossectional Study

PONE-D-21-34897R2

Dear Dr. Habte Hailegebireal,

We’re pleased to inform you that your manuscript has been judged scientifically suitable for publication and will be formally accepted for publication once it meets all outstanding technical requirements.

Kind regards,

Saqlain Raza

Academic Editor

PLOS ONE
---

## [Editor Report · Acceptance letter]

17 Oct 2022

PONE-D-21-34897R2 

The Uptake of Key Essential Nutrition Action (ENA) Messages and Its Predictors among Mothers of Children aged 6-24 months in Southern Ethiopia, 2021: A Community-Based Crossectional Study 

Dear Dr. Habte Hailegebireal:

I'm pleased to inform you that your manuscript has been deemed suitable for publication in PLOS ONE. Congratulations! Your manuscript is now with our production department. 

Kind regards, 

on behalf of

Dr. Saqlain Raza 

Academic Editor

PLOS ONE